# Tissutal and Fluidic Aspects in Osteopathic Manual Therapy: A Narrative Review

**DOI:** 10.3390/healthcare10061014

**Published:** 2022-05-31

**Authors:** Marco Verzella, Erika Affede, Luca Di Pietrantonio, Vincenzo Cozzolino, Luca Cicchitti

**Affiliations:** Accademia Italiana Osteopatia Tradizionale, 65127 Pescara, Italy; marco.verzella@aiot.edu (M.V.); affede.erika@gmail.com (E.A.); vincenzo.cozzolino@aiot.edu (V.C.); cicchittiluca@gmail.com (L.C.)

**Keywords:** exclusion zone water, interstitial fluid pressure, water, somatic dysfunction, osteopathic manipulative treatment, low-grade inflammation, antidromic activity, fibroblasts

## Abstract

Over the years, several authors have discussed the possibility of considering somatic dysfunction (SD) as a “nosological element” detectable on palpation. There are many aspects to consider regarding the etiology and diagnosis of SD, and the literature on osteopathic issues provides details on physiological signs that characterize it, including tissue texture changes. Recent knowledge suggests that how tissue and, in particular, connective tissue, responds to osteopathic treatment may depend on the modulation of the inflammation degree. Low-grade inflammation (LGI) may act on the extracellular matrix (ECM) and on cellular elements; and these mechanisms may be mediated by biological water. With its molecules organized in structures called exclusion zones (EZ), water could explain the functioning of both healthy and injured tissues, and how they can respond to osteopathic treatment with possible EZ normalization as a result. The relationship between inflammation and DS and the mechanisms involved are described by several authors; however, this review suggests a new model relating to the characteristics of DS and to its clinical implications by linking to LGI. Tissue alterations detectable by osteopathic palpation would be mediated by body fluids and in particular by biological water which has well-defined biophysical characteristics. Research in this area is certainly still to be explored, but our suggestion seems plausible to explain many dynamics related to osteopathic treatment. We believe that this could open up a fascinating scenario of therapeutic possibilities and knowledge in the future.

## 1. Introduction

The main means available to osteopathic medicine is to assess tissues by palpating, in particular tissues of the musculoskeletal system, with the aim of diagnosing a possible SD.

By underscoring some contradictory aspects, several authors have called into question SD, by defining it as a nosological entity detectable on palpation [1,2,3].

SD is classified by the ICD 11 [4] as a “Biomechanical lesion, not elsewhere classified”; however, the definitions are not equally shared and codified by osteopathic professionals [1,5,6,7].

SD presents the characteristics of impaired or altered function of components related to the somatic system, involving skeletal, arthrodial, and myofascial structures, and osteopathic manipulative treatment (OMT) is aimed at the treatment of SD [8,9,10].

The osteopathic literature describes the relationship between SD and OMT in many studies [10,11,12,13].

OMT is a drug-free manual medicine, a patient-centered, whole-body intervention. OMT has shown positive effects in different fields such as gynecology and obstetrics, neonatology, chronic inflammatory disease management, and musculoskeletal disorders [14,15,16,17,18,19].

There are many aspects to consider regarding the etiology and diagnosis of SD, and the osteopathic literature provides details on the signs that characterize it, including tissue texture changes [8,20,21,22].

Over the last few years, some authors have proposed a variety of interpretation models in order to clarify the mechanisms of onset and the inherent characteristics of tissue alterations concerning SD. Among such models, there are also clinical reasoning and decision-making procedures suitable to establish a treatment routine [23,24,25,26,27].

Recent knowledge suggests that tissue, and, in particular, connective tissue, may react by modulating the inflammation degree. This issue should also be extended to any response to OMT, and several studies show the efficacy of OMT on inflammatory tissue levels [28,29,30,31,32,33,34,35].

LGI would act on the ECM, and alter its structure, such as in fibrosis, which is defined as a lesion of the connective component in an organ or tissue [36].

These alterations occur through mechanisms mediated by the environment in which the tissues are placed, namely water [37,38,39,40,41].

The water under consideration is water present in living matter. It has particular biophysical characteristics, which could exemplify the functioning of both healthy and injured tissues [39,42,43].

This review suggests a new model regarding the characteristics of SD and its clinical implications, comparing it with LGI, whose tissue alterations would be mediated by biological water located near the membranes [39,40].

## 2. Methods

We performed the following narrative review following the guidelines of the Gasparian et al. study [44].

The research on literature was carried out between September and December 2020, by using the following databases such as MEDLINE (https://pubmed.ncbi.nlm.nih.gov/), (accessed on 30 December 2020) SCOPUS (https://www.elsevier.com/solutions/scopus) (accessed on 30 December 2020 ), Scholar Google (https://scholar.google.com/) (accessed on 30 December 2020), and the Cochrane Library (https://www.cochranelibrary.com/) (accessed on 30 December 2020).

The investigation was performed by means of the following keywords: “Fibroblasts” [MeSH Terms], “Myofibroblasts” “ [MeSH Terms], “Fascia” [MeSH Terms], “Interstitial Fluid Pressure” [Free Terms], “Dipole Waves” [Free Terms], “Exclusion-Zone Water” [Free Terms], “Flow Sensing” [Free Terms], “Water” [MeSH Terms], “Dissipative Structure” [Free Terms], “Shear Stress” [MeSH Terms], “Mechanosensors” [Free Terms], “Interstitial Fluid Flow” [Free Terms], “Aquaporin” [Free Terms], “Inflammation” [MeSH Terms], “Antidromic Activity” [Free Terms], “Low Grade Inflammation” [Free Terms], “Metabolic Diseases” [Free Terms], and “Para-Inflammation” [Free Terms].

For the search strategy, we used Bolean operators as “AND” and “NOT” and all the key terms alone. Then we matched the results.

### Inclusion Criteria of the Papers

All the studies involving both human and animal testing, as well as laboratory (in vivo and ex vivo) studies were included.

The search strategy included reviews, clinical trials, and observational studies. All the other kinds of studies were excluded.

Any kind of restrictions regarding publication date and study outcome were set and only the studies written in English were considered.

Additional research was also performed through the reference list of the included articles, resulting in a “snowball procedure” [45]. 

All the duplicates were identified and removed due to the computer software Zotero, and the online software of the Corporation for Digital Scholarship.

Three members of this research group (EA, LDP, and MV) carried out a preliminary search independently of one another, producing 13,730 results in a total amount. After excluding duplicates, 11.331 articles were removed. Any discrepancies were resolved by consensus with LC as referee. A selection method was performed by dividing the whole process into three consequent steps: (1) title screening; (2) abstract screening; and (3) full-paper screening.

## 3. Results

Seventy-five articles were included in this review (Figure 1). In the interest of clarity, it was decided to divide the results into 3 macro topics titled: (1) redefine inflammation; (2) some water, cells, and body fluids; and (3) biophysics aspects.

### 3.1. Redefine Inflammation

In living organisms, inflammation is the adaptive and defensive mechanism against a large number of harmful stimuli by modulating tissue repair in order to restore physiological functions [46,47].

Recent findings on inflammation show that particular pathologies are mediated by inflammatory responses defined as LGI or “chronic low-grade inflammation” [48,49,50,51,52]. Medzhitov [37] suggests that certain chronic pathologies, such as type 2 diabetes and cardiovascular diseases, and, in particular, atherosclerosis could not be caused by the well-known mechanism of inflammation. LGI creates a different degree of expression of cellular and, consequently, tissue function, thus generating a *malfunction.* As a result, the tissue adaptive response is modulated by the quantitative state of inflammation. The author defines this condition with the term *para-inflammation,* defining it as a phenomenon that stands between the homeostatic basal state of the tissue and the actual inflammatory response.

This response would be mediated mainly by the macrophages residing in tissues; therefore, the function of para-inflammation would be to stimulate the tissue to adapt to stress conditions. Para-inflammation would be present without any obvious infection or injury, and its sustained status for prolonged periods would advance the tissue into a condition of chronic inflammation.

To confirm the role and characteristics of LGI, Antonelli et al. [52] specify that the dynamics of this type of inflammation differ from a normal inflammatory condition. In the latter, there is a high concentration of the elements of the innate immune response (inflammatory cytokines), associated with high levels of C-reactive protein (CRP) (Figure 2A).

In the case of LGI, such conditions would not be present. In fact, CRP levels prove to be modest, so neither is the result of infections or tissue lesions, nor the fundamentals of inflammation defined by Celso, such as heat (calor), pain (dolor), redness (rubor), and swelling (tumor), are present. Therefore, LGI would be a systemic condition linked to the alteration of the functions of tissues and cells that deviate from a state of homeostasis, a phenomenon that often increases over time [37].

The concept of homeostasis, replaced in recent years by the term allostasis, is intended as a system of adaptation to environmental variations, “stability through change” [38,53,54]. The allostatic load represents the metabolic expenditure to preserve such an adaptation.

If this load is excessive and lasts for a long time, it is capable of generating a condition of exhaustion of adaptive capacities (allostatic overload), a possible cause of potentially severe pathologies. The substances involved are inflammatory and anti-inflammatory circulating cytokines, glucocorticoids, and catecholamines [55,56].

Tissue dysfunctions would be triggered not only by macrophages, but also by dendritic cells and by a variety of cells that perform the homeostatic monitoring of tissues.

According to a recent study [52], when changes in the internal environment lead to cellular stress (metabolic stress, injury, and pathogens), LGI manifests itself as an innate immune response.

At the base of cellular and, therefore, tissue stress, with consequent activation of inflammation, there are mechanisms that can lead to the development of some pathologies, including unfolded protein response (UPR) [57]. This cellular response to environmental and metabolic insults (e.g., cytokines, glucose deprivation, altered cellular redox status, hypoxia, viruses, increased protein trafficking, excess or deficiency of certain nutrients) [58] would disrupt protein folding and the accumulation of proteins in the endoplasmic reticulum (ER), thus leading to cell apoptosis [59,60].

Relating to allostatic overload, an important role is played by psychosocial stress such as work overload, unemployment, or caring for a family member with a life-threatening chronic disease [38,61,62].

According to Rohleder et al. [61], there is more experimental evidence underscoring a direct relationship between psychosocial stress and LGI. In fact, there is an increase in cytokines as well as intracellular activity suitable for inflammatory signaling.

The effect of tissue chronic inflammation is frequently showed by its structure alterations, such as fibrosis, defined as a lesion of the connective component in an organ or tissue, a consequence of the increase in the fibrillar portion of the extracellular matrix (ECM). This condition can occur in a variety of vascular, metabolic, and tumor pathologies. Sclerosis, on the other hand, occurs at a later phase of fibrosis; it is commonly associated with the increase of the consistency and the hardness of the fibrotic tissue if the alterations of the ECM persist, and it is macroscopically appreciable also by palpation [36].

Tissue alterations are conditioned by matrix metalloproteinases (MMP), a class of enzymes produced by resident macrophages, which are activated in the turnover of ECM and would be influenced by some types of interleukins (IL-13 and IL-4) [63,64].

In particular, the lack of balance between MMP and its inhibitor, mediated precisely by inflammatory conditions, would largely contribute to the development of fibrosis in the tissues by increasing the deposition of the fibrillar component [36,65,66].

Ultimately, LGI is the result of a series of causative elements: continuous and recurrent noxious stimuli, metabolic alterations, and, last but not least, environmental and social stress which, with different physiopathological dynamics, would determine tissue alteration [49,50].

### 3.2. Some Water, Cells, and Body Fluids

Connective tissue represents the main site of inflammatory processes [67] and is an extensive network closely connected to the external environment both through cellular contiguity and ECM [41]. It is also considered the determining key for the transmission of mechanical forces, which, in turn, influence pathological and physiological processes, from wound healing to inflammation, and even cancer [68,69].

Fibroblasts (FB) and myofibroblasts (MFB) are mainly responsible for the tension of the matrix, as well as for its stiffness and viscosity [70]. Connective stiffness, and, in particular, of the fascia, has been studied by several authors. Some argue that cellular contractility is influenced by the sympathetic nervous system, considering the response of MFBs to TGF-β [71]; others believe that it is influenced by the expression of different cytokines within the matrix and by the pH level of the matrix itself [72].

In the study by Schleip et al., the tissue stiffening effect is not caused solely by the active cell contraction of FB/MFB, but by the change in the hydration of the matrix. Indeed, water constitutes the major component of the volume of the fascia and different stresses on it affect the speed of rehydration [73].

A recent study on the interstitium supports the postulation of the reciprocal influence among mechanical forces, fluid dynamics, and cellular response. The collagen bundles constituting the complex network of the interstitium are in fact devoid of a basement membrane and, therefore, directly in contact with interstitial fluid (IF) [74].

Therefore, what follows is a direct interaction between fluid forces and the remodeling of the local fiber capable of providing the cell with sensitive mechanical feedback with respect to the architecture of the matrix [75].

Fluid flows inevitably create fluid shear stress (FSS) [75] which, in turn, acts as a regulator of different biological processes (differentiation and gene expression) on multiple cell types: FB/MFB, including pluripotent and somatic stem cells [76], according to the principles of mechanotransduction [77,78,79].

Furthermore, IF can alter the extracellular distribution of chemokines, or secreted morphogens, and, thus, direct cell migration or capillary morphogenesis, as well as the alignment of local ECM fibers [80].

Fluid activity in the biophysical environment of tissue inflammation is also important in the absence of exogenous mediators, such as TGF-β1, and with low levels of interstitial flow [81]. Fluid dynamics, therefore, has significant implications for the tissue, both from a functional and pathological point of view [74].

Inflammatory mechanisms determine an effect on the migration and accumulation of fluids in the extracellular space [37] and a condition of increased cell swelling; these are all signs of an altered homeostasis that is expressed by edema, whose mediators would be IF and its pressure (IFP). In fact, the recall of fluids from the vascular stream towards the interstitial space, at least in the initial phase of the edema, would depend on a reduction in the IFP, a mechanism that is involved in various inflammatory reactions and tissue trauma conditions [82,83].

At the base of the lowering of IFP, there would be the release of the tensions exerted by fibroblasts on the collagen networks and microfibrils in connective tissue.

The molecular mechanism would be determined by the blocking action of cytokines (PGE1, IL-1, IL-6, and TNF) on the membrane integrins of fibroblasts, causing a loss of tension in ECM, with subsequent fluid recall. Only in the later stages of the edema, would fluid recall be due to the increase in capillary permeability and hydrostatic pressure [68,84].

It is interesting to note that integrin blocking by proinflammatory cytokines, and, consequently, the extent of IFP lowering, is mediated by the degree of inflammation and can also occur with LGI. The extracellular matrix would act like a sponge which, losing its tension capacity, swells, and drawing fluid inside it [68]. In cases of significant IFP reduction due to high inflammation and capillary filtration, it increases up to 10–20 times, and is associated with collagen denaturation, which also occurs in the case of neurogenic inflammation [84] (Figure 2B).

As it pertains to changes in cell fluids in response to inflammation, an important role would be played by aquaporins (AQP) [85,86]. AQPs are a group of membrane channel proteins that facilitate the passive transport of water within the cell. They are present in large numbers in a variety of body tissues (brain, synovium, and cartilage), and also expressed on the membranes of cells that do not play an evident role in the transport of fluids (adipocytes and muscle cells) [87,88].

In the presence of processes associated with changes in cell volume, such as migration, inflammation, proliferation, and cell death, AQPs have shown considerable importance, and many authors agree in defining them as real regulators of osmotic inflammation induced by stress [89,90].

It can be suggested that the morphological and, therefore, functional alterations of tissues are due to the quantity and displacement of fluids within them and mediated by an inflammatory condition.

### 3.3. Biophysics Aspects

In light of what has just been described and taking into account that water represents the main component of living matter (about 70% of the total mass and 99% of the number of molecules), we must broaden our understanding of the dynamics relating to tissues, by considering biological water as a fundamental element with discrete behaviors [42].

In the proximity of hydrophilic surfaces, and, in particular, of biological membranes, the molecules organize in a distinct order in large areas of the water mass volume. These zones are described as exclusion zone (EZ) or “vicinal water”, so defined because particles and solutes measuring from 10 to 0.1 nanometers are rejected in this thickness [43,91,92,93].

EZs have a distinctive water structure, in which molecules are organized in honeycomb layers, which overlap in a parallel direction with the membrane surfaces, both inside and outside the cell [94]. EZ water has specific chemical/physical properties. It has higher viscosity and is more stable than “bulk” water (distant from hydrophilic surfaces): its molecular motility is more restricted; its light absorption spectrum is greater than bulk water, both in UV and IR rays range; and, finally, it has a higher refractive index [43].

Proteins within living organisms are surrounded by water organized in EZ [95]. When this water is lacking, the proteins would be outside their *normal* functional environment, causing an anomaly in their folding.

Most cellular functions, such as muscle contraction, secretion, and nerve conduction, depend on protein folding [57,58,59,60]. Water organizes in EZ also in the stabilization of the triple helix of collagen and in the orientation of mineral particles within the bone matrix [96]; EZ alteration would thus lead to impaired or pathological functions [39,43].

According to the authors, a possible explanation for the holistic effects of a variety of health-promoting agents and substances, such as nutraceuticals and certain types of fats, could lie in their ability to restore the accumulation of EZ water within cells, thus influencing general health [39,40].

Water EZs, therefore, play a role of considerable importance both at an extracellular and intracellular level; Kerch [97] associates aging and various pathologies to the presence of “loosely bound water”.

EZs would also have effects on fluid movement, since, within tubular structures, such as blood capillaries, fluids can move independently of the pressure gradient. This occurs due to a potential difference between the nucleus of the tubule and the internal surface of EZs, creating a flow that the authors call “self-driven flow” [98,99].

The same authors hypothesize a scenario in which blood flow is possible in some districts without cardiac contraction. More in detail, sulfate molecules, which are found in most of the cells of the body and, in particular, in the endothelium, would play a role in preserving EZs.

The negative charge provided by sulfate ions attached to glycosaminoglycans in the capillary wall generates an electromagnetic field called electrokinetic vascular streaming potential (EVSP) [96,99]. This determines not only the ease of movement of the fluid due to a buffer effect, but also a reaction of the endothelium itself, promoting the release of nitric oxide [100].

One of the theoretical models used to analyze and explain the interactions and organization of biological water is that of quantum field theory (QFT) and quantum electrodynamics (QED) [101,102]. According to these models, molecules must be considered as objects that can inherently promote spontaneous fluctuation, which can exchange energy with the surrounding environment. In suitable environmental conditions, such as the density of molecules and the right temperature, all units spontaneously oscillate in unison in accordance with a well-defined phase and in synchrony with the electromagnetic field (EMF), which has the same phase [42,101].

This spatial region is called the *coherence domain* (CD), which has a submicron size and contains many millions of molecules [42,102].

In the water EZ, the molecules organize themselves, forming CDs. A phase block takes place within them, giving rise to a collective molecular oscillation constituted by the wavelength of EMF [42,93,101,102,103,104]. In biological tissues, the presence of electrons in the water available engenders the possibility of generating chemical reactions. In fact, the coherent oscillation of water molecules occurs between the minimum energy configuration, in which all electrons are strongly bound, and an excited configuration whose energy is just below the level necessary to tear an electron from the molecule (12, 06 eV) [105,106,107].

Thus, the biological water in the EZ is *nearly free of* electrons available for metabolic chemical reactions, while remaining bound to the water molecule [93]. The water CDs receive energy from the external environment and the light, which would increase the potential energy of the EZ with reserve functions [108] (Figure 2C).

In living organisms, CDs can organize themselves into a coherent set of several CDs, which would bring the spatial extension of the coherent region up to macroscopic sizes, such as those of cells, organs, and tissues [42,101].

From a thermodynamic point of view, water CDs can be considered as dissipative structures. A dissipative system is able to self-organize through the continuous exchange of energy with the external environment, decreasing its entropy [106,109,110,111].

In fact, when the CD oscillation frequency corresponds to that of some types of target molecules, (non-aqueous monomers), found at its outer limits, these molecules are attracted by the EMF of the CD, becoming part of it [107].

Molecules using the energy stored in the CD can thus perform chemical reactions that produce new species of biomolecules, determining the lowering of entropy and leading to an evolution over time of the biological organism [107,112]. Ultimately it would be water, with its CDs and their EMFs, providing the opportunity to different chemical reactions to take place, with precision, between some molecules and not with others [113].

## 4. Discussion and Hypothesis

With the acronym TART (tenderness, asymmetry, range of motion abnormality, and tissue texture changes), osteopathic literature accurately provides the characteristic elements of SD, at which OMT is aimed [7,8,21]. However, some authors disagree on the relevance to be attributed to different clinical signs: some indicate the range of motion abnormality as fundamental for a diagnosis of SD, but there is no univocal evidence on the reproducibility in the evaluation [114]. Other authors suggest the need for the presence of at least 2 of these 4 signs; still, others do not consider the sign of hypersensitivity or tenderness [1,5,6,7,115]. Regarding the asymmetries of the musculoskeletal structures, these can occur for a variety of causes, and are, therefore, difficult to attribute solely to SD [116,117,118].

In light of the results of this review, we believe that among the 4 clinical signs considered, tissue texture changes are the most significant to define an SD, thus proposing the hypothesis that SD can be compared to a condition of LGI.

The mechanisms underlying SD are still widely discussed in the literature, but it is reasonable to think that without first having tissue texture changes, caused by inflammatory phenomena, the presence of the other three clinical signs is not possible.

We suggest that an inflammatory phenomenon could determine an alteration of the tissue as described in the chapters above, and only subsequently tenderness, altered movement, and asymmetry of the musculoskeletal structures can occur (Figure 2D).

The timing just described could be explained by one of the most accredited mechanisms of the onset and maintenance of SD: the neurogenic inflammation [1,119,120], in which the primary afferent nociceptors (PAN) determine the release in the periphery of neuropeptides, such as substance P and calcitonin gene-related peptide (CGRP). The neurotransmitters mentioned above are released into the peripheral peri-vascular and extracellular space through an antidromic signal, causing a local inflammatory response with alterations of the surrounding tissue. It should be noted that this area, by means of the axonal branch, can be very large [121].

These neuropeptides have vasoactive functions, recalling immune cells, activating mast cells, and releasing histamine, thus acting on the trophic state of the innervated organ [21,119,121,122]. Together, they contribute to the possible genesis of tissue alterations, also influencing the recovery of tissue lesions and their repair [123].

The nerve fibers involved would be predominantly the poorly myelinated C or A-delta, fibers of the interoceptive component which, therefore, represent the afferent portion of the sympathetic efference [124].

It has been demonstrated that sympathetic efference plays a decisive role in the onset of inflammatory phenomena [125]. These findings agree with what Denslow and Korr underscored regarding SD, as it pertains to expressiveness of phenomena related to neurogenic inflammation [126] and autonomic sympathetic innervation [127].

There may be mechanisms capable of leading to tissue alteration, which are associated with the dynamics of neurogenic inflammation. These dynamics are all probably linked to inflammatory phenomena, such as the unfolded protein response (UPR) [57,58,59,60], as well as the alteration of the functions of the MMP [63,64], which would determine changes to the functions of the ECM [65,66]. Last but not least, the allostatic overload would cause tissue alteration [61].

SD does not represent a real pathological condition [4]. In fact, as for LGI, it would not have a direct cause, such as trauma or tissue injury. Rather, SD appears as an alteration in tissue function, a sign of altered homeostasis, often lasting over time, and, like LGI, it can be placed between a homeostatic basal state and the actual inflammatory response [37,52].

There are studies on the efficacy of OMT in healthy people diagnosed with non-symptomatic SD [115,128,129,130]; the clinical conditions of these subjects could be associated with LGI, in which the blood inflammation markers are modest.

However, the signs of DS are not associated with the classic signs of inflammation. SD represents a sign of metabolic alteration that manifests itself with the alteration of the tissue texture, leading to tissue fibrosis and possible sclerosis and, therefore, is diagnosable through palpation [20,21,36].

The existence of a restriction barrier within the range of motion, a characteristic sign of SD [20,21,22], implies the alteration, both quantitative and qualitative, of a tissue or a joint region in a given district. This alteration is generated on an inflammatory substrate, without necessarily showing signs of classic inflammation [22].

### Hypothesis

Once tissue alterations and their cause linked to LGI phenomena have been confirmed, we hypothesize that the mediating element is the particular behavior of fluids in the tissues. In fact, tissue stiffness would not be caused exclusively by the active contraction of the cellular populations present, for example, of fibroblasts in the connective tissue, but it could also derive from the modification of the water amount of the ECM [73].

The mode of distribution of fluids within our organism should be redefined taking into account the new findings on the characteristics of the interstitium and its topographical location in the various body districts [74,131].

The same forces developed by the movement of fluids, such as FSS and IFP, relative to an LGI, can determine alterations in the function and shape of the tissue itself [68,75,76,80,81,84]. At the cellular level, AQP-mediated fluid shifts could play a predominant role in tissue alterations, taking into account that these membrane channels respond to inflammatory phenomena [85,86,88,89,90]. Biological water would react to a series of mechanisms related to inflammation that could be explained by biophysics. The presence of EZ would be crucial and its absence would determine real dysfunctions of cells and tissues, leading to a pathological condition [39,132].

The displacement of fluids would therefore not be linked only to anatomy, as considered up to now, but could be dependent on the structure of the EZ in vessels and capillaries [96,98,99,100]. This mechanism could also intervene at the level of the interstitium and in larger areas of the body [131] with greater freedom of fluid movement.

It can be hypothesized that the responses to OMT at the tissue level, verified during and immediately after treatment, may depend on the fluid dynamics within the ECM, with mechanisms still to be elucidated. In fact, in the short term, these effects would not be consistent with the timing relating to matrix remodeling. If this were the case, a relatively longer period would be required [70].

As shown by the results of in vitro studies, the restoration of tissue functions, and, in particular, of the connective tissue through indirect osteopathic techniques [133], could result from the production of growth factors and anti-inflammatory cytokines (e.g., platelet-derived growth factor-bb). These would be able to reverse the block of activity on integrins, restoring normal tension and, therefore, hydration of the ECM [28,29,73,84,133,134].

However, there is no robust evidence of OMT efficacy on inflammatory diseases in the osteopathic literature [16,35,128,135]. This could depend on the reliance on reference parameters that are hardly indicative: LGI and, therefore, SD do not have a high level of inflammatory markers detectable in routine blood tests.

Inflammation could be considered a process necessary to restore the conditions of tissue stability [46]. The resulting tissue turnover would be the means by which the sulfates, present in glycosaminoglycans and necessary for the existence of EZ, are re-synthesized. The result would be the restoration of a physiological EZ of the water that is adequate for metabolic processes [96].

Considering that the state of tissue health can be linked to the presence of water, it is conceivable that the effect of OMT lies precisely on the restoration and normalization of EZs, in areas with macroscopic coherence domains, through the intervention of EMFs of the tissue itself.

This hypothesis, to be fully verified experimentally, would find support in the statements of De Ninno et al., who believe that it is possible to establish electromagnetic homeostasis (EH) related to living tissues that would allow a much faster and long-distance intercellular communication [93].

According to our hypothesis, the anti-inflammatory action of OMT would determine an effect on the ECM and on the restoration of the optimal electromagnetic conditions (through the sulfates on the cell membranes) to reconstitute the EZ on the cell membranes and then on the tissues, a condition defined as electromagnetic homeostasis (Figure 2E).

The loss of the collective oscillation of CD, related to the reduction of the thickness of EZs, would not allow the biological tissue to dissipate energy with the external environment. This condition would increase its entropy, thus making it lose the ability to self-organize [42,93,107] and, consequently, its health.

The results described align with the knowledge of the fundamental elements of osteopathic medicine, such as the primary respiratory mechanism (PRM) with its 5 principles, and the dynamics of fluids related to osteopathic treatment [136,137].

## 5. Conclusions

Our review shows the importance of the role of biological fluids on health and their relative behavior in the osteopathic approach.

Results in this review highlighted a different vision of SD by comparing it to LGI, this allows us to define the tissues texture changes as the main signs to be considered in the diagnosis of SD.

Manual therapy, in particular indirect osteopathic techniques, would have effects on multiple biological areas, probably linked to the regulation of water EZs and to the inflammatory state of the tissues. By restoring the adequate basic fluidic substrate in a biological system, osteopathic therapy would allow the manifestation of self-healing mechanisms.

This principle is consistent with the historical principles of osteopathy, which the present review attempts to expand with the support of recent literature. Research in this area is certainly still to be deepened, but our proposal seems plausible to us in order to frame the many dynamics relating to osteopathic treatment. We believe that this could open up a fascinating scenario of therapeutic possibilities and knowledge in the future.

## Figures and Tables

**Figure 1 healthcare-10-01014-f001:**
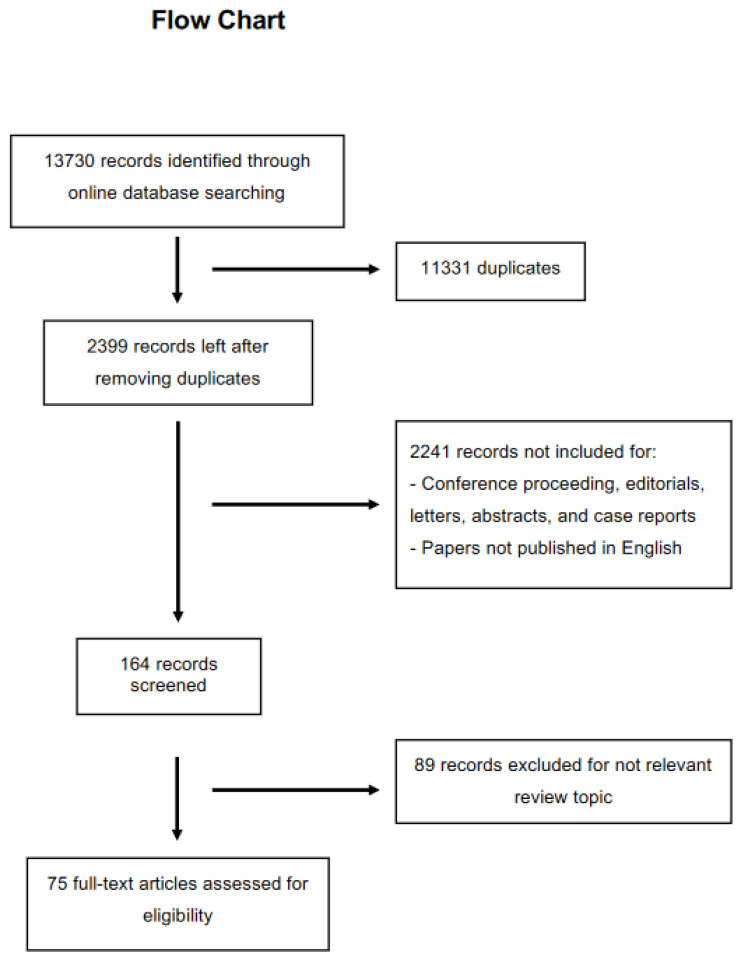
Flow chart of the study selection.

**Figure 2 healthcare-10-01014-f002:**
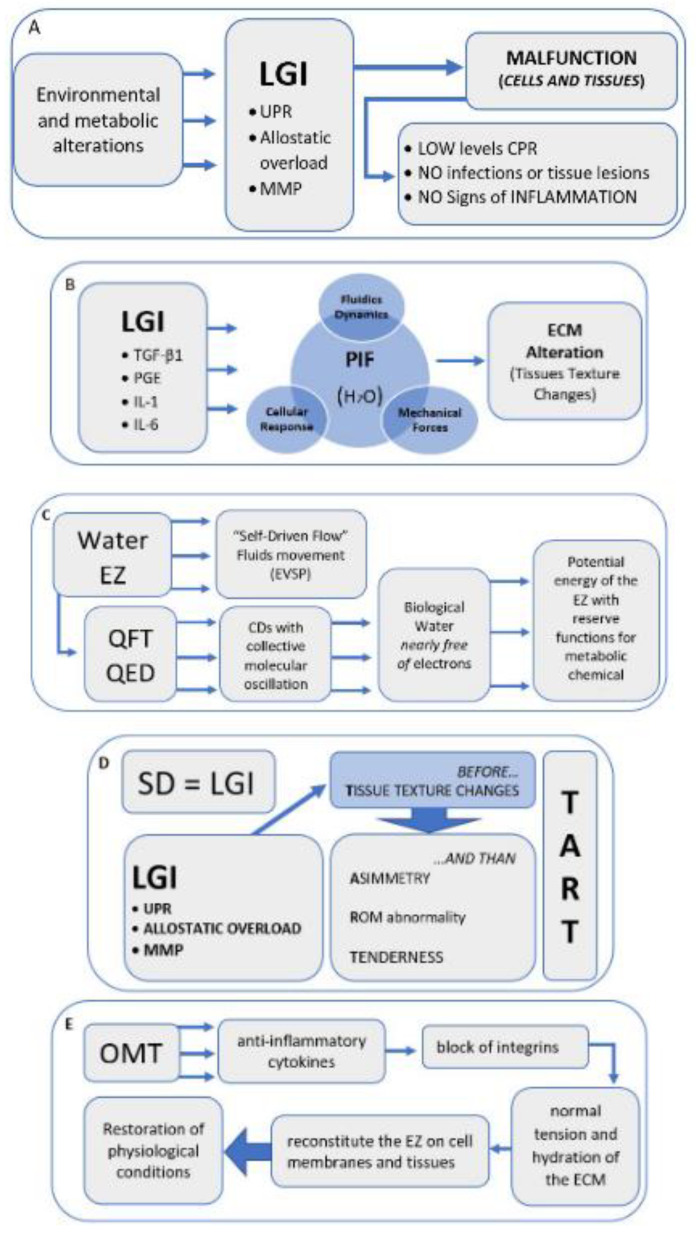
Narrative flow charts.

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
