# Peer review of "Tissutal and Fluidic Aspects in Osteopathic Manual Therapy: A Narrative Review"

_healthcare, 2022, doi:10.3390/healthcare10061014_

Round 1

Reviewer 1 Report

Marco et al present a manuscript providing a comprehensive review on the role of biological fluids on health, with relative behavior subordinated to the osteopathic approach. The authors also addressed osteopathic therapy have effects on multiple biological areas, probably linked to the regulation of water EZs and to the inflammatory state of the tissues.

However, the paper needs very significant improvement before acceptance for publication. Detailed comments are as following:

1. I do not recommend publication of the manuscript in healthcare in its present state and I find that the authors could improve the manuscript in that direction. For the moment the manuscript is incomprehensible and directionless. I found this passage unconvincing. I think the relationship between on the Somatic Dysfuntion and Osteopathic Manipulative Treatment should be explained better.

2.The abbreviations should be consistent throughout the article or list the complete form when the first use of abbreviations. For example, in the “Introduction” section, ICD 11 abbreviation has spelled out when it is first used in the “Introduction” section. And in the “Abstract collection” section, Low-grade inflammation abbreviation has spelled out,afterwards only the abbreviation should be used. please check if all abbreviations.

3. The title of the manuscript is ambiguous. Please simplify the title to better reflect the core findings of the manuscript. Also, the introduction reads like part of a review article-it broadly surveys the field rather than leading the narrative to the specific questions that will be discussed in the word.

4. Abbstract need to be somewhat better, at the moment significance of this work andtissue aspects in Osteopathic Manual Therapy is not really presented and needs to rewritten to show the strength of presented results and their application.

5. This manuscript needs much better justification why we need this results/science, and what is the significance of this work. How we can benefit from it? Who need that?

6. The article draws some conclusions, but both the writing and the representation of experiments make it sometimes difficult to follow the logic. The authors should explain how interstitial fluid pressure, low grade inflammation, antidromic activity, and fibroblasts to relieve tissue aspects can respond to osteopathic treatment. And the authors should have in-depth discussion about the priority of discussion about osteopathic treatment in this review, the advantages of osteopathic treatment compared to other therapy should be discussed.

Reviewer 2 Report

The authors have covered the topic “Role of water in the Osteopathic Manual Therapy” in this review. They have extensively discussed the Inflammation, the role of water, and the biophysics behind it. They also have made a hypothesis based on their narration. The information here would be beneficial for OMTs. However, there are a few concerns in the current version

  1. Figure 1 delivers no information regarding the topic. I suggest the authors replace this figure – add a narrative flow chart explaining subsections in results and their connection to the main objective of the topic.
  2. Section 3.2 could use a narrative figure to describe the process.
  3. Sections 3.2 and 3.3 reads like an introduction to the hypothesis. Authors should consider deepening some of the concepts in sections 3.2 and 3.3.
  4. Cells’ plasticity also affects their morphological changes. Authors should consider discussing this in section 3.3
  5. The authors have covered many domains in this review. However, the language is not easing the transition from one domain to another. Hence, I recommend that the authors consider working on the transition for readers from different backgrounds.

Reviewer 4 Report

The authors aimed to suggest new interpretative approaches concerning the characteristics of Somatic Dysfunction (SD) and its clinical implications, also proposing a new paradigm.

The study covers some issues that have been overlooked in other similar topics. The structure of the manuscript appears adequate and well divided in the sections. Moreover, the study is easy to follow, but few issues should be improved. Some of the comments that would improve the overall quality of the study are:

  1. Authors must pay attention to the technical terms acronyms they used in the text. Please better stated the aim of the study in the abstract section.
  2. English language needs to be revised.
  3. Conclusion Section: This paragraph required a general revision to eliminate redundant sentences and to add some "take-home message".

Round 2

Reviewer 3 Report

The text is improved in the latest revision.

There are still minor changes to be made.
